



# Mesoscale Convective Systems as a source of electromagnetic signals registered by ground-based system and DEMETER satellite

Karol Martynski[1], Jan Blecki[2], Roman Wronowski[2], Andrzej Kulak[1], Janusz Mlynarczyk[1], Rafal Iwanski[3]

[1]Department of Electronics, AGH University of Science and Technology, Kraków, Poland
[2]*Space Research Centre PAS, Warsaw, Poland*
[3]*Satellite Remote Sensing Department Institute of Meteorology and Water Management - National Research Institute Cracow, Poland*

*Correspondence to*: Karol Martynski (karol.martynski@agh.edu.pl)

**Abstract.** Mesoscale Convective Systems (MCS) are especially visible in the summertime, when there is an advection of warm maritime air from the West. Advection of air masses is enriched by water vapour, which source can be found over the Mediterranean Sea. In propitious atmospheric conditions, thus significant convection, atmospheric instability or strong vertical thermal gradient, lead to the development of strong thunderstorm systems. In this paper we discuss one case of MCS, which generated a significant amount of +CG, -CG and IC discharges. We have focused on the ELF (Extremely Low Frequencies, < 1 kHz) electromagnetic field measurements, since they allow to compute the charge moments of atmospheric discharges. Identification of the MCS is a complex process, due to many variables, which have to be taken into account. For our research we took into consideration a few tools, such as cloud reflectivity, atmospheric soundings and data provided by the PERUN system, which operates in VHF range (113.5 – 114.5 MHz). Combining described above measurement systems and tools we identified a MCS, which occurred in Poland on 23 July 2009. Furthermore it fulfilled our requirements since the thunderstorm crossed the path of DEMETER overpass.

## 1. Introduction

Mesoscale Convective Systems (MCS) are enormous cloud structures known as one of the strongest discharge generators in the world (Bonner, 1968; Banta et al., 2002; Houze, 2014). In the past many have focused on the topic of MCS activity e.g. Price (2002) or Cummer (2004). In our previous work (Martynski et al., 2018) we analysed a supercell that occurred over Poland. In this paper our main goal is to combine measurements from two autonomous systems, the ELF (Extremely Low Frequency) Hylaty station and DEMETER satellite that measures ELF and VLF (Very Low Frequencies) fields. Additionally, we used PERUN (Polish system of the discharge localisation system) for tracking the storm cells and to scrutinise individual lightning discharges.

To conduct the analysis we have designated one event of the MCS which occurred on 23 July 2009. The selection of this specific MCS was not random, we have been looking for the most favourable conditions to conduct the





research, hence significant convection, atmospheric instability or strong vertical thermal gradient. Since we focused on Poland we had access to data provided by the Polish meteorological service IMGW-PIB (Institute of Meteorology and Water Management – National Research Institute). The most crucial part in the search for MCS was to scrutinise DEMETER overpasses in order to distinguish periods, when it was over Poland.

## 35  2. Detection of electromagnetic signals generated by atmospheric discharges

Ground-based measurements were conducted by the Hylaty ELF station (Kulak et al. 2014), localized in the Bieszczady mountains in Poland. The station measures the electromagnetic field in the ELF range, we used a receiver that is operating in 0.03 – 55 Hz frequency range and uses two orthogonal magnetic antennas, one aligned with North-South and the second with East-West. Based on these measurements, inverted solutions developed within our team had been used, which support

computation of the charge moments of CG discharges (Kulak et al., 2010). This method requires two parameters, one is the distance between the discharge and the receiver and the second is the amplitude of the recorded impulse. The MCS was located approximately 350 to 450 km from the station, and was heading North-East.

DEMETER has been operating till December 2010 and had sun-synchronus orbit. The ELF/VLF range for the electric and magnetic fields is from DC up to 20 kHz. There are two scientific modes: a survey mode where spectra of one

electric and one magnetic components are on-board computing up to 20 kHz. Time resolution for the spectra in this case is about 2s and frequency resolution about 19 Hz. The second one is a burst mode, where in addition to the on-board computed spectra, waveforms of 3 electric and 3 magnetic field components are recorded with sampling up to 2.5 kHz. During the burst mode the waveforms of the six components of electromagnetic field were registered with 2550 Hz sampling rate. It allows to perform a spectral analysis in the range up to 1250 Hz with much higher time and frequency resolution (0.4 ms and

0.8 mHz) (Parrot et al., 2006; Berthelier et al., 2006). Since the burst mode is used occasionally we had to distinguish periods where the measurements are corresponding with thunderstorms. As mentioned above the satellite allows to measure signals in VLF and ELF ranges, which aids the search of these relations (Parrot et al. 2008; Parrot et al. 2013; Blecki et al. 2016). Thus differences and variations in the ionosphere and magnetosphere, caused by thunderstorm activity (Berthelier et al., 2006).

## 55  3. Results

From the South-West, Poland was covered by a warm tropical air masses. Their advection over colder polar maritime air caused the occurrence of a significant thermal contrasts between Western and Eastern Europe. For instance temperature difference in Benelux and Eastern Germany and Poland, was larger than 15°C. Furthermore due to temperature discrepancy in tropospheric layers a jet-stream had occurred in the middle troposphere (700 hPa) with the air flow around 15-25 m/s.

Thus conditions were favourable for wind-shear to occur, which is vital for thunderstorm development. Over Southern Sweden, a local low is noticeable, that caused flow of the cold front from the Western Europe. Around 19 UTC over the Lower Silesia (Polish voivodeship) a derecho had occurred, well known for strong wind gusts often exceeding 40 m/s, the





thunderstorm's path covered an area of length of 1000 km and width varying between 50 to 200 km. Lastly high thermodynamic instability lead to the updrafts. Discussed example of MCS is qualified as derecho, due to the propitious

synoptic conditions and the destruction it caused (Evans and Doswell, 2001).

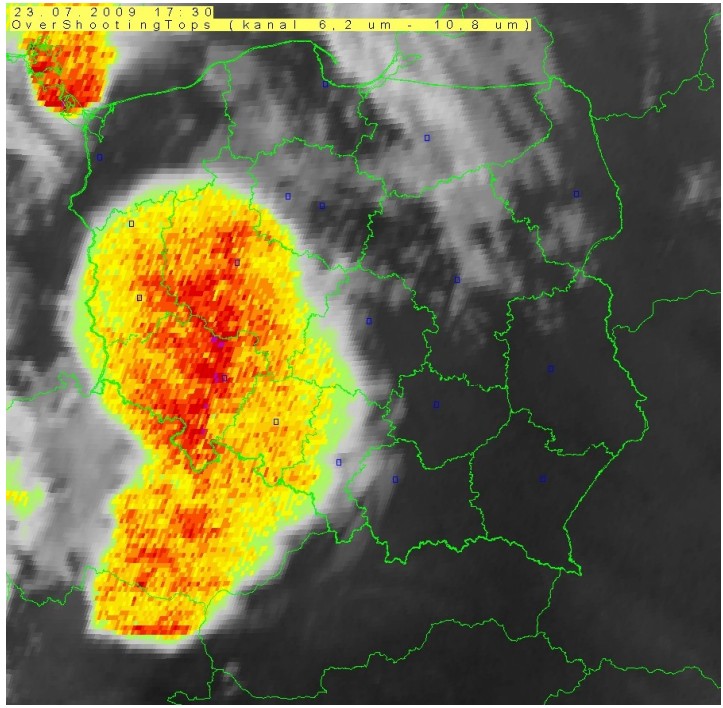




**Figure 1** Satellite WV-IR OST product (Overshooting Tops) – 17:30 UTC - 23 July 2009.

        Vertical wind shears support the separation of the updrafts from downdrafts, moreover they support processes that are responsible for development of the multicellular thunderstorms. Discussed air flow in the middle troposphere

allowed the whole system to move with relatively high velocity. The convergence in the lower troposphere let the Bow Echo to form, which manifested as a squall line. Last, but not least, another significant condition for MCS development is the advection of the cold air mass from the Western regions of Europe. Thermodynamic conditions, which appeared over South-Western Poland, additionally confirm development of the strong convective phenomena. The data from atmospheric soundings show high temperature level at the ground layer (over 30°C), with dew temperature at 22°C. A strong air flow

from the West is visible in the whole troposphere. Thermodynamic indicators such as CAPE (Convective Available Potential Energy) 2500 J/kg or CIN (Convective Inhibition) -100 J/kg indicates strong convective processes. A significant drying of air appears, then a dry adiabatic gradient is noticeable in the middle troposphere. Small inversion layer (CIN) favours "the gathering" of the energy beneath it, when convection is strong enough it is possible to break through the inversion, which





directly leads to the intensification of the convection processes. Then tropopause is penetrated by the convection and an overshooting top may appear in the lower parts of the stratosphere (Fig. 1). Apart from the discussed thermodynamic parameters, SBCAPE (Surface Based Convective Available Potential Energy) is significant. The parameter indicates the convection in the surface layer, in this case it exceeded 2500 J/kg. Furthermore DCAPE (Downward Convective Available Potential Energy) is available, which is the potential of downdrafts – 1077 J/kg. Wind shear parameter in 0-6 km was higher

than 20 m/s, whereas in 0-3 km the parameter was equal to 13 m/s. A significant development of the thunderstorm phenomenon is visible by the measurements of the cloud reflectivity. In many parts of the MCS a level that exceeds 50 dBZ is distinguishable that indicates strong convective processes, which supplied cloud development at the level higher than 15 km over the ground level.

Strong atmospheric discharges stem from significant MCS development. In the period of the highest

thunderstorm activity 24 +CG (Cloud-to-Ground), 322 -CG and 2836 IC (Intracloud) discharges were detected (Fig. 2). Additionally we provide data for other periods where a significant amount of discharges occurred (Table. 1), which indicates an enormous extent of IC discharges. The intensification of the discharges with highest wind gusts is visible in the front parts of MCS.


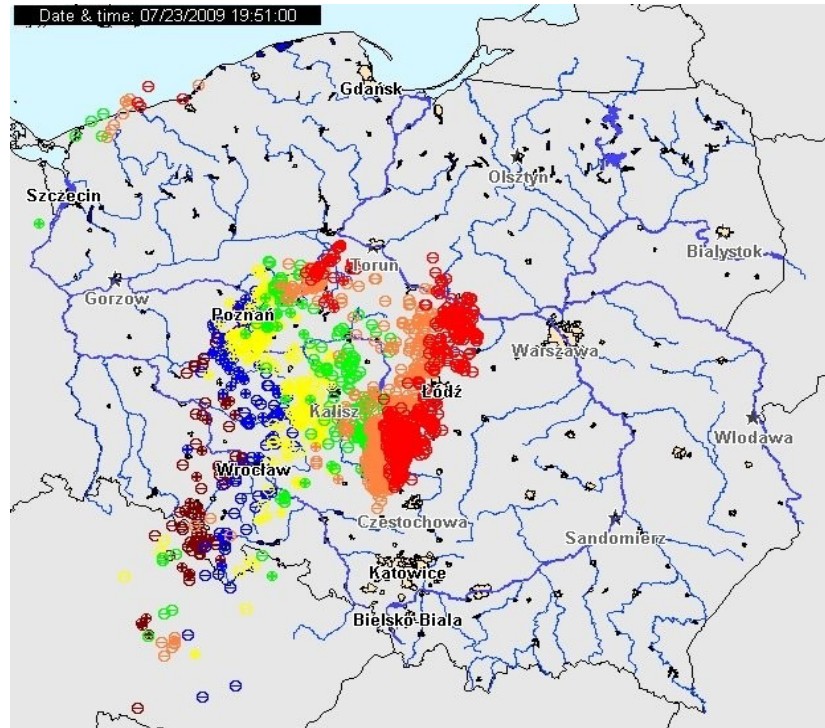



**Figure 2** Map of the atmospheric discharges for the most active period (+/-CG and IC) based on PERUN system - 23 July 2009.

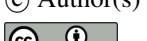



**Table 1.** Number of +CG, -CG and IC strokes during the MCS activity -  23July 2009 - data received from the PERUN system.

| Time | 16:20 | 17:20 | 18:20 | 19:20 | 20:20 |
|------|-------|-------|-------|-------|-------|
| **+CG** | 2 | 6 | 9 | 5 | 2 |
| **-CG** | 5 | 10 | 14 | 148 | 45 |
| **IC** | 320 | 865 | 240 | 917 | 494 |

During the analysis of the DEMETER data for the whole lifespan of the discussed MCS, we have encountered a signature  of a whistler - a characteristic type of waves that occurs in VLF frequency range.  The whistlers are a cold plasma waves in the frequency range from the ion cyclotron up to the electron plasma frequency or electron cyclotron frequency. These waves are common in space around Earth and may be registered in the ionosphere and the magnetosphere, by the satellite onboard receivers as well as by the ground-based systems. The characteristic shape of whistler's spectrum with falling frequency in time is a result of its dispersion feature and propagation. The group velocity is greater for waves with higher frequencies than for lower ones. The whistlers propagate along magnetic field lines from the site of the thunderstorm. The arrival of the lower frequency waves is delayed in relation to higher frequency (Helliwell, 1965; Hayakawa, 1995).

Figure 3 presents a whistler that has been detected by DEMETER overpass, which was 287 km away from the causative lightning stroke. Its precise location  was provided by PERUN.  During that time an impulse was caught by ground-based systems, which detected an impulse slightly ahead than the satellite, PERUN detected a signal at 20:05:49.13 and classified it as a +CG with the maximum current of 24 kA. Hylaty measurements distinguish an impulse at 20:05:49.14 with an amplitude 220 pT and a charge moment of 103 C km. The satellite registered a signal at 20:05:49.23 with an electric field 1200 µV/m the magnetic field is omitted due to high noise.  The whole period of MCS activity has been presented in Figure 4.  Data presented in the figure represents a distribution of the charge moments, which were computed from data collected by the Hylaty ELF station. As a result, only lightning with the charge moment above 23 C km is included in our analysis The highest charge  moment during MCS lifespan was 328.9 C  km. Total amount  of +CG discharges, which occurred was  1073. The average value of charge moment is 42.5 C km.






**Figure 3** Comparison of DEMETER and Hylaty results. Top panel presents the waveform of the electric field in the ELF range, the middle panel shows a spectrogram of the electric field in VLF range both registered by DEMETER and the bottom panel presents the magnetic field in the ELF range in time interval 20:05:47-20:05:51 - 23 July 2009.




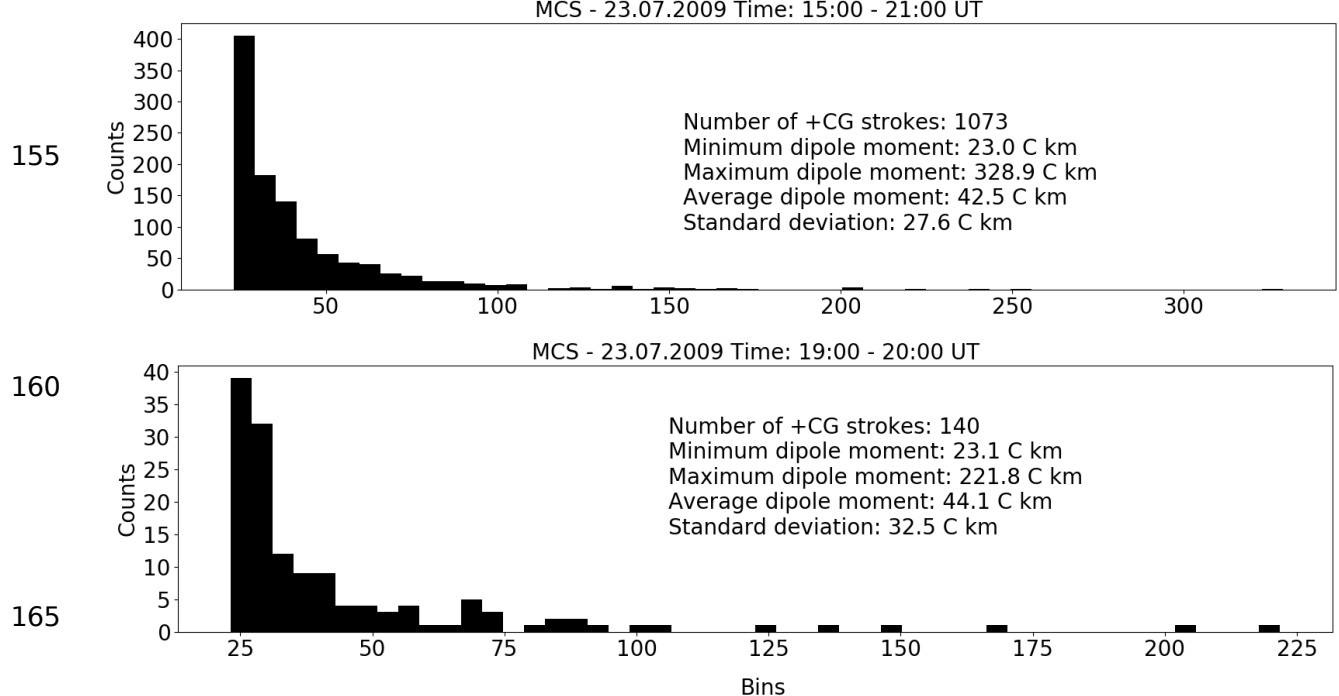




**Figure 4** Histograms of charge moment distribution in the analysed MCS: for the whole activity period (top) and for the
most intensive hour (bottom).

## 4. Summary

In this paper, we focused on a specific type of thunderstorms, an MCS classified as a derecho, which is not frequent in
Europe and may be disastrous for overpassed areas. It is known mostly for the intense wind gusts and small amount of
discharges. Comparing results from our previous paper about supercell activity (Martynski et al., 2018), we can conclude
that supercell in the most active hour generated almost equal amount of discharges to the MCS case during its whole life
cycle. That indicates that the supercell generated at least two times more discharges than MCS, although the strongest
strokes are still produced by the huge cloud clusters. This means that MCS are more developed storm cells, although they
are unable to generate multiple strong discharges. This might be due to a fact that they cover up larger areas comparing to
supercells, which structure is more organised and condensed, allowing heat and storm-like processes to develop as a stronger
singular thunderstorm. Similar results to ours have been obtained by Cummer (2004), who stated that MCS on High Plains in
the USA generated in one hour roughly 312 +CG strokes and the mean charge moment was equal to 36.8 C km. In this study
we also presented a whistler that was registered by DEMETER during its overpass of the MCS. This shows the potential of
combined ground-based and satellite studies .



## Data availability


Data may be obtained by request sent to the main author of this paper.

## Competing interests

The authors declare that they have no conflict of interests.

## Authors contribution

Karol Martyński provided conceptualization, data curation, formal analysis, investigation, project administration, software,
writing to the publication. Jan Błęcki and Roman Wronowski allowed funding acquisition, conceptualization and
investigation. Janusz Młynarczyk and Andrzej Kułak created a methodology, software and data curation for this paper. Rafał
Iwański was responsible for data visualisation, data curation and resources. All above authors were supervisors of the paper
since the topic is broad and requires expertise in many fields.


## Acknowledgements

The studies were conducted with financial help the National Science Centre, grant No. 2017/27/B/ST10/02285

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
