# Peer review of "Mesoscale Convective Systems as a source of electromagnetic signals registered by ground-based system and DEMETER satellite"

_Annales Geophysicae, 2020_

## Referee Comment (RC1) · Anonymous Referee #1 · 13 Dec 2020

I think this is a good paper, with detailed analysis of some specific event of MCS. The authors bring together data from ground based registrations of ULF/ELF waves and response in the ionosphere in ELF/VLF range with meteorological data on MCS and lightning activity. The week side of this paper is analysis of only one case, but the analysis is performed very well and the paper can be treated as case study and worth to be published.

For non-specialist It would be useful to add photo or drawing of the MCS. It would improve brightness of the presentation.

Authors use data from space instruments not being their own they should acknowledge

of the DEMETER PI,s of the instruments which data have been used.

I recommend to publish this paper after these minor improvements.

---

## Referee Comment (RC2) · Anonymous Referee #2 · 18 Dec 2020

The paper on Mesoscale Convective Systems (MCS) by Martynski et al. is of good quality and desserves to be published in Annales Geophysicae. The analysis concerns new data, both from satelite and ground measurements. Appart from analysising the large-scale MCS systems and the electric discharges associated with them (strong discharges were detected), the Authors have also observed a whistler in their data, which I think of interest to potential readers. Moreover, the conditions leading to formation of MCS were analysied in detail, which was particularly impressive.

---

## Referee Comment (RC3) · Anonymous Referee #1 · 22 Dec 2020

Thanks for the clarification regarding the MCS photo. I think the manuscript is ready for publication.

---

## Short Comment (SC1) · 22 Dec 2020

I'm thankful for your comment, I would like to answer for your concerns with this paper. First, below the email I'm attaching acknowledgements for DEMETER team, as suggested.

Secondly, we have shown only one case of MCS, since it was difficult to match DEMETER flybys with such a rare phenomenon. These appear rather rarely in Poland, thus the material for the paper was sparse. A photo of the MCS is redundant, since these phenomena are recognised mostly by the scrutinised analysis, rather than a photography. Furthermore these types of thunderstorms are a complex cloud formations and

might be difficult to distinguish them only by "taking a look", because they share some morphological features with other types of thunderstorms, such as supercells.

Acknowledgements. The studies were conducted with financial support of the National Science Centre, grant No. 2017/27/B/ST10/02285. We express our gratitude to M. Parrot, J-A. Sauvaud, J-J. Berthelier, J-P. Lebreton, PI's of DEMETER instruments from which data were used.

Kind regards Karol Martyński

---

## Short Comment (SC2) · 24 Dec 2020

Thank you for your time and for the review.

Kind regards Karol Martyński
* * *

---

## Author Comment (AC1) · 31 Jan 2021

We would like to thank the reviewer for taking the time to review our paper and pointing out where it can be improved. We added an acknowledgments for DEMETER team, as suggested. We have shown only one case of MCS, because it was difficult to match DEMETER flybys with its occurrence. Mesoscale Convective Systems rarely appear in Poland, thus the material for the paper was sparse. It would be difficult to distinguish it in a photo because they share some morphological features with other types of thunderstorms, such as supercells these phenomena are recognised mostly by the scrutinised analysis, rather than a photography.

---

## Author Response (AR1)

Due to the first reviewer comment, we've added „We express our gratitude to M. Parrot, J-A. Sauvaud, J-J. Berthelier, J-P. Lebreton, PI's of DEMETER instruments from which data were used." to **Acknowledgements** section.